# Comparative In Vitro and In Silico Enzyme Inhibitory Screening of *Rosa x damascena* and *Pelargonium graveolens* Essential Oils and Geraniol

**DOI:** 10.3390/plants12183296

**Published:** 2023-09-18

**Authors:** Ayşe Esra Karadağ, Sevde Nur Biltekin, Betül Demirci, Fatih Demirci, Usman Ghani

**Affiliations:** 1Department of Pharmacognosy, School of Pharmacy, Istanbul Medipol University, 34815 Istanbul, Türkiye; 2Department of Pharmaceutical Microbiology, School of Pharmacy, Istanbul Medipol University, 34815 Istanbul, Türkiye; snbiltekin@medipol.edu.tr; 3Department of Pharmacognosy, Faculty of Pharmacy, Anadolu University, 26470 Eskişehir, Türkiye; bdemirca@anadolu.edu.tr (B.D.); fdemirci@anadolu.edu.tr (F.D.); 4Faculty of Pharmacy, Eastern Mediterranean University, North Cyprus, 99450 Famagusta, Türkiye; 5Clinical Biochemistry Unit, Department of Pathology, College of Medicine, King Saud University, Riyadh 12372, Saudi Arabia

**Keywords:** angiotensin converting enzyme II, cholinesterase, cyclooxygenase, lipoxygenase, cytotoxicity

## Abstract

The present work aims to evaluate *Rosa x damascena* Herrm. and *Pelargonium graveolens* L’Hér. essential oils, and the major constituent geraniol for their in vitro and in silico inhibitory activities against *5*-lipoxygenase (5-LOX), cyclooxygenase (COX), acetyl cholinesterase (AChE), butyryl cholinesterase (BuChE), and angiotensin converting enzyme (ACE2) enzymes. Geraniol most potently inhibited the ACE2 relative to other enzymes. *R. damascena* essential oil moderately inhibited the cancer cell lines with no toxic effects on healthy HEK 293 cells. *P. graveolens* essential oil inhibited a number of cancer cell lines including A549, MCF7, PC3, and HEK 293 that are reported here for the first time. The molecular docking of geraniol with the target enzymes revealed that it binds to the active sites similar to that of known drugs. Geraniol carries the potential for further drug development due to its drug-like binding mode for the target enzymes. Our work confirms that these essential oils possess similar biological activities due to their similar phytochemistry in terms of the major constituents of the plants. The promising biological activities reported in this work further warrant the inclusion of in vivo studies to establish safe use of the target essential oils and their constituents.

## 1. Introduction

It is well known that the *Rosa x damascena* Herrm. essential oil is widely used in the aromatherapy, cosmetics, and food industries along with other utilizations. *R. damascena* is a natural hybrid of *R. gallica* L. and *R. phoenicia* Boiss. from the Rosaceae family. It has been reported to ethnobotanically treat inflammatory conditions and mental disorders. *R. damascena* flower decoctions are traditionally used to relieve chest and abdominal pain, menstruation pain, digestive system disorders, and constipation, and to strengthen the heart. In addition to the traditional use of rose water as an antiseptic for eye washing and as a mouth disinfectant, it is also used as a muscle relaxant to relieve abdominal pain, and bronchial and chest congestion [1,2]. Its flowers, petals, and fruits have been traditionally used to treat insomnia in Iran [2]. *R. damascena* flowers are reported to be traditionally used as a laxative, diuretic, and refreshment, as well as for the relief of ailments such as general weakness, cold, cough, flu, abdominal pain, and gallstones [3]. Rose oil is used to heal depression, sadness, nervous stress, and blood pressure problems [4,5]. It is used for the reduction in prolonged coughing and gynecological complaints, including the healing of wounds and the treatment of skin diseases [5]. Treatment involving rose oil vapors has been reported to be beneficial for some allergies, headaches, and migraines [4,5]. In previous experimental studies, its antimicrobial effect was reported against various human pathogens [6]. Additionally, the anticonvulsant, neuroprotective, and bronchodilator effects of *R. damascena* essential oil have been reported in previous studies [4,5,6,7].

It is known that *P. graveolens* essential oils are used or adulterated instead of *R. damascena* oil due to the similarity of its chemical profile, smell, and perception. *P. graveolens* of the Geraniaceae family is an important aromatic plant. To date, more than 120 components have been identified in its essential oil, the most important of which are citronellol and geraniol. The essential oil is valuable since it is used in soaps, perfumery, aromatherapy, and the cosmetics industry [8]. It is also commonly used as a flavoring agent in foods due to its strong rose-like smell. *P. graveolens* is traditionally used for skin problems, and for the relief of pain, coughs, colds, sinusitis, and tuberculosis [8]. The antioxidant, hypoglycemic, antimicrobial, and antifungal effects of *P. graveolens* essential oil have also been reported in previous studies, along with other uses [9,10].

Enzymes are promising targets for the treatment of various diseases and conditions associated with their function in the body. The design and development of clinically important enzyme inhibitors as drugs are important steps toward the treatment of diseases. The present work targets a number of such enzymes that are highly implicated in the development of various diseases. Among them is cyclooxygenase (COX) that is responsible for the biosynthesis of mammalian prostaglandins from arachidonic acid. It is a known target for the treatment of pain and inflammation. Moreover, it is also implicated in the development of certain forms of cancer [11]. The isoform COX-2 is a membrane-bound enzyme, which is also overexpressed in many types of cancer, promoting carcinogenesis and increasing cancer cell resistance to chemotherapy as well as to radiotherapy [12].

Angiotensin-converting enzyme 2 (ACE2), a zinc metallopeptidase, is the only known human homolog of the enzyme [13]. It is mainly associated with heart function, hypertension, and diabetes. ACE2 is an exopeptidase that catalyzes the conversion of angiotensin 2 to angiotensin 1–7 and *L*-phenylalanine [13]. ACE2 is a type-I integral membrane glycoprotein that acts as a carboxypeptidase rather than as a dipeptidase [14]. The main locations of the receptors of this enzyme, which is active and expressed in most tissues, are cells in external contact, such as enterocytes of the small intestine and alveolar epithelial cells of the lungs. ACE2 is also found in venous and arterial cells, renal and cardiovascular tissue, and smooth muscle cells [15]. ACE2 is also one of the target receptors for SARS-CoV, the human respiratory coronavirus NL63, and the novel coronavirus 2019 nCoV/SARS-CoV-2. Previous studies have also reported that ACE2 is one of the essential receptors for various coronaviruses’ entry into cells through the spike protein [16].

The mammalian cholinesterase neurotransmitters acetylcholinesterase (AChE) and butyryl choline (BuChE) and their inhibitors represent the most promising strategy for potential treatment of Alzheimer’s disease since the accumulation of β-amyloid plaques is controlled by both enzymes. It has been shown that the deposition of β-amyloid plaques in the brain is strongly correlated with AChE and BChE activities. Inhibition of the enzymes is an important clinical target that elevates the level of acetylcholine–butyrylcholine in the brain, which in turn reduces β amyloid plaque aggregation in the brain [17].

Since the essential oils of *R. damascena* and *P. graveolens* may be phytochemically similar and can be interchangeably used due to their odor similarity, some biological activities of the oils and their major compounds, in particular geraniol, are investigated in the present study. Therefore, the purpose of the current work is to test the hypothesis that safe traditional uses of the phytochemically similar essential oils, especially geraniol, translate to similar biological activities in vitro and in silico.

Based on the uses of *R. damascena* and *P. graveolens*, the present work comparatively evaluates commercial *R. damascene* and *P. graveolens* essential oils, and the major constituent geraniol for their in vitro and in silico anti-inflammatory, anticholinesterase, potential antiviral, and cytotoxic activities.

## 2. Result and Discussion

### 2.1. Phytochemical Analysis

The *R. damascena* and *P. graveolens* essential oils were acquired from commercial sources and analyzed to confirm their compositions and quality before in vitro biological evaluations. The oil compositions were reported according to gas chromatography–flame ionization detector (GC-FID) and gas chromatography–mass spectroscopy (GC/MS) analyses as relative percentages (%), which are listed in Table 1, with the total sum of 96 and 98.9%, respectively. Overall, 44 components were identified where the major component of *R. damascena* and *P. graveolens* essential oils was 27.2% and 38.7% geraniol, respectively. Furthermore, *R. damascena* essential oil was confirmed to contain 23.5% citronellol, 16.4% nonadecane, 10% nerol, 6.8% heneicosane, and 2.2% germacrene D, whereas the major components other than geraniol for the *P. graveolens* essential oil were identified as 19.3% citronellol, 6.3% citronellyl acetate, 5.7% 10-epi-δ-eudesmol, 5.6% isomenthone, 5.4% linalool, and 3.2% geranyl formate (see Table 1 for details).

According to previous studies [4,5,18,19], the phytochemical profile of *R. damascena* essential oil was found to be similar to the results obtained in the present work. Likewise, *P. graveolens* essential oil contained abundant amounts of geraniol and citronellol as also reported in previous work [20].

### 2.2. Enzyme Inhibition and Computational Studies

#### 2.2.1. COX and LOX Enzyme Inhibition and Geraniol Binding Modes

Geraniol interacted with the active sites of each enzyme mainly through hydrogen bonding and hydrophobic interactions. Its only hydroxyl group is indispensable for inhibiting the target enzymes. The binding energies of geraniol for all enzymes are listed in Table 2.

The IC_50_ values of the essential oils and geraniol for COX-1 and COX-2 inhibition were calculated using a range of concentrations. *R. damascena* and *P. graveolens* essential oils and geraniol showed IC_50_ values of 24.4, 11.1, and 11 µg/mL for COX-1, whereas for COX-2 and 5-LOX, the values were 39.1, 50, and 8.1, and 19.1, 23.1, and 7 µg/mL, respectively (Table 3).

Since COX-2 selectivity is an important parameter for COX enzyme inhibition, our data supported the selectivity index (SI) as shown in Table 4. The essential oils were more selective to COX-1 compared to other enzymes. However, unlike the parent essential oils, geraniol selectively inhibited COX-2. The results showed that *R. damascena* essential oil is effective for both COX enzymes, whereas *P. graveolens* essential oil was ineffective to COX-2 inhibition at the tested concentrations. Geraniol significantly inhibited both COX-1 and COX-2 enzymes. Both essential oils and geraniol were also more effective to 5-LOX inhibition.

Past work on *P. graveolens* essential oil has also shown that it is more selective to COX-1 [21]. Although the anti-inflammatory effect of *R. damascena* essential oil has already been reported in the literature [22,23], our work further extends to comparative biological and in silico studies on geraniol and geraniol-rich *P. graveolens* essential oil.

Geraniol binds within the active sites of COX-1 and COX-2 in a manner similar to that of non-steroidal anti-inflammatory drugs (NSAIDs), including indomethacin ethanolamides [24], fenamates [25], and celecoxib and its derivatives [26]. It involves the main active site residues of both enzymes for interaction that is comparable to that of the above drugs. In the COX-1 active site, the hydroxyl group of geraniol shares two hydrogen bonds, one each with Tyr 385 and Ser 530 amino acids (Figure 1a). These two residues are catalytic and critical for the cyclooxygenase reaction cycle [27]. Other residues including Ala 527, Ile 523, Phe 518, Val 349, and Trp 387 also play key roles in conferring specific orientations to geraniol through hydrophobic interactions that facilitates hydrogen bonding with the catalytic residues.

Geraniol also binds within the cyclooxygenase channel of human COX-2; yet again, its hydroxyl group actively shares two hydrogen bonds with the side chains of Arg 120 and Tyr 355 residues. It does not interact with the catalytic Tyr 385 and Ser 530 residues, since the entire molecule is rotated 180° away from them, facilitating hydrogen bonding interactions with Arg 120 and Tyr 355 residues on the other side instead (Figure 1b). Geraniol’s hydrophobic contacts with Phe 381, Leu 384, Tyr 385, Val 523, and Ala 527 residues, located within the catalytic site, substantially contribute to its further stabilization in the COX-2 active site. The binding mode of geraniol for COX-2 is significantly like that of COX-1, involving many common active site residues that are conserved in both enzymes. A comparison of the molecular interactions of geraniol in the active sites of COX-1 and COX-2 is presented in Figure 2. The general position of geraniol in the active sites of both COX-1 and COX-2 is significantly similar despite differences in its molecular conformations and amino acid interactions with both enzymes. The ability of geraniol to inhibit both enzymes indicates it to be a promising candidate for designing dual inhibitors of COX-1 and COX-2 enzymes.

5-LOX is an Fe^2+^-containing enzyme that catalyzes the conversion of polyunsaturated fatty acids, particularly arachidonic acid and linoleic acid, to hydroperoxides for the synthesis of leukotriene inflammatory mediators. Leukotrienes are implicated in the pathogenesis of various diseases and pathological processes such as atherosclerosis, Alzheimer’s disease, dementia, cerebral ischemia, asthma, and inflammation [28]. Targeted inhibition of leukotriene synthesis through the inhibition of 5-LOX is an important therapeutic approach to abating inflammatory processes in the cell, and for treating the above diseases and processes. Common 5-LOX inhibitors used as drugs for the treatment of asthma and inflammation include meclofenamates, benzothiophenylethyl hydroxyurea, and natural products such as nordihydroguaiaretic acid (NDGA) and acetyl-11-keto-beta-boswellic acid (AKBA) [29]. The structural biology of some of these inhibitor complexes indicate their binding in the active site cavity that also involves the catalytic zinc center coordinated by His 518, 523, and 709 residues [30]. Geraniol binds in the same cavity as the above drugs where its hydroxyl group interacts with one of the coordinating histidines (His 518) through a hydrogen bond. It is otherwise stabilized in the pocket by several hydrophobic residues that constitute the cavity, which include Leu 565, Ile 572, Thr 575, Phe 576, Val 569, Ile770, and Leu 773 (Figure 3). The binding modes of 5-LOX drugs share similarities with that of geraniol that may be of interest in designing new and more effective 5-LOX inhibitors as drugs.

In different animal models, both *R. damascena* extracts and essential oil have been mechanistically investigated for anti-inflammatory and analgesic effects [31,32]. Furthermore, geraniol was also reported to possess anti-inflammatory activity using animal models, therefore further validating the anti-inflammatory effects of *R. damascene* essential oil [33]. We propose that geraniol has a major role in the anti-inflammatory effect.

#### 2.2.2. ACE2 Enzyme Inhibition and Geraniol Binding Mode

The *R. damascena* and *P. graveolens* essential oils at a concentration of 20 µg/mL inhibited ACE2 at 56.4% and 67%, respectively. The ACE2 enzyme inhibition by geraniol alone at 5 µg/mL was 86.3%. These results indicate that the essential oils were not significantly effective against ACE2; however, it is geraniol that exhibits major inhibition of the enzyme. Geraniol may be a potential inhibitor of SARS viral entry in the cells such as SARS-CoV2 since it inhibits ACE2. 

ACE2 carboxypeptidase is a membrane protein that plays an essential role in the regulation of heart function. It is also a functional receptor for the cellular entry of coronaviruses, causing severe acute respiratory syndrome (SARS) including COVID-19. The structure of ACE2 is divided into two subdomains, namely subdomain I and II. The catalytic center is surrounded by S1 and S1′ subsites for substrate or inhibitor binding. The two sides of the active site are formed by subdomains I and II. The former is a zinc-containing subdomain constituted by 19–20, 290–397, and 417–430 amino acid residues, whereas the latter is formed by 103–289, 398–416, and 431–615 residues [34]. Geraniol binds to the S1 site in subdomain II away from the zinc-containing subdomain I mainly through four amino acid residues (Figure 4). The Arg 273 and His 505 residues are crucial for geraniol binding that establish three hydrogen bonds with it. The two side-chain amino groups of Arg 273 form two hydrogen bonds with the hydroxyl group of geraniol. Additionally, the His 505 sidechain forms a hydrogen bond with the hydroxyl group of geraniol through its imidazole ring amino group. Geraniol is further stabilized through hydrophobic interactions with Phe 274 and Thr 276 residues. The phenyl ring of Phe 274 mainly stabilizes the remaining hydrocarbon structure of geraniol through hydrophobic contacts. Moreover, there is also a secondary role of Thr 276 that anchors the terminal carbon of the inhibitor via hydrophobic interaction.

Our work also potentially establishes a clinical correlation between the ethnobotanical use [3,4,8] of *R. damascena* and *P. graveolens* plants for treating flu and colds with ACE2 inhibition since the enzyme is highly implicated in upper respiratory viral infections. As geraniol is the major constituent of the essential oils, it is reasonable to target it for the development of antiviral drugs. 

#### 2.2.3. AChE and BuChE Enzyme Inhibition and Geraniol Binding Modes

Table 4 lists the IC_50_ values of *R. damascene* and *P. graveolens* essential oil along with geraniol for AChE inhibition.

In Ottoman traditional medicine and modern aromatherapy, the use of *R. damascena* essential oil is known to increase focus and attention and improve memory [35]. Geraniol may be one of the active compounds responsible for such effects since previous studies supported the memory-enhancing effects of *R. damascene* essential oil in animal models [36].

The AChE and BuChE are important enzymes for synaptic neurotransmission in the central and peripheral nervous systems. They are strongly implicated in the development and progression of Alzheimer’s disease. Inhibition of the enzymes reduces the degradation of acetylcholine that increases its exposure at the synapses, thereby improving neurotransmission in Alzheimer’s and other neurological diseases [37].

The active sites of AChE and BuChE are composed of a deep substrate-binding gorge surrounded by the acyl- and choline-binding pockets as in Figure 5. The gorge is common in both AChE and BuChE for many species [37]. The catalytic triad of the active site is located near the bottom of the gorge. The main amino acid residues of the AChE gorge include Asp 74; Thr 83; Trp 86; Gly 120-122; Tyr 124, 133, 337, and 341; Trp 286; Phe 295; and 297 with the catalytic triad formed by Glu 202, Ser 203, and His 447 residues. The BuChE gorge comprises Asn 68, Asp 70, Trp 82 and 231, Ala 328, Phe 329, and Tyr 332 residues with the catalytic triad constituted by Glu 325, Ser 198, and His 438 residues. The gorge is also present in the electric eel AChE active site, but with differences in amino acid residues compared to the human enzyme. Current drugs such as donepezil, huperzine, and galantamine are known to specifically bind in this gorge [38]. The binding mode of geraniol shares similarity with that of the drugs mentioned above. The entrance to the gorge is on the surface of both enzymes that forms an open channel to the inner cavity, where geraniol settles and stabilizes itself (shown in Figure 5).

This type of binding is unlike that of donepezil, which almost occupies the whole area of the gorge from the entrance to the bottom due to its long and bulky molecular structure [38]. Geraniol engages several AChE amino acid residues for hydrogen bonding and hydrophobic interactions. Its hydroxyl group simultaneously forms two hydrogen bonds with the Tyr 133 side-chain hydroxyl group and with one of the amino groups of the catalytic Glu 202 side chain. The rest of the interactions involve hydrophobic contacts with the side chains of TRP 86, Tyr 124 and 337, and Phe 297 and 338 residues (Figure 6a). Geraniols’ interaction with the catalytic Glu 202 residue indicates that it may be a mechanism-based inhibitor of AChE.

Geraniol displays a different binding mode in the active site of BuChE where it establishes a hydrogen bond with the amino group of the Trp 82 side-chain ring with additional stabilization provided by hydrophobic contacts with Trp 82, Tyr 128, Trp 430, and Tyr 440 residues, as illustrated in Figure 6b. Due to its binding mode akin to known drugs, the potential of geraniol can be further explored for designing new AChE and BuChE inhibitors for the treatment of neurological diseases.

### 2.3. Cytotoxic Activity

The IC_50_ values for *R. damascene* and *P. graveolens* essential oils and geraniol on A549, MCF7, and PC3 cells are listed in Table 5. The essential oil showed no cytotoxic effects on HEK 293 cells after 24 h of exposure. 

*R. damascena* essential oil showed a relatively low cytotoxicity against the target cancer cells. The highest cytotoxic effect on the MCF7 cells was observed at a concentration of 105 µg/mL. However, no cytotoxic effect was detected on the healthy cell line, even at the maximum concentration of 2 mg/mL. The results showed that *P. graveolens* essential oil was more effective than that of *R. damascena* in terms of cytotoxicity on both cancerous and healthy cells. Additionally, *P. graveolens* essential oil also showed the highest cytotoxic effects on MCF7 cells similar to that of *R. damascena* essential oil. Geraniol showed the most cytotoxic effects on MCF7 cells. On a comparative basis, the lowest cytotoxicity was detected on PC3 cells, as shown in Table 5.

*R. damascena* essential oil has been demonstrated to be effective both in the vapor phase and with direct contact application against different cancer cells [39]. Although there are reports on the cytotoxic effects of the plant extract and rose water [40], to the best of our knowledge, this is our first report on the anticancer and cytotoxicity activity of *R. damascena* essential oil.

The cytotoxic effects of *P. graveolens* essential oil on MCF7 cells have been previously reported [21]. To the best of our knowledge, this is the first report on the effects of the essential and geraniol on A549, PC3, and HEK293 cells.

In general, our work justifies the correlation of the traditional use of Rose and Pelargonium essential oils for different ailments. Both essential oils showed a remarkably selective inhibitory activity against the COX-1 enzyme, and apoptosis against selected cancer cell lines. Molecular docking on geraniol, especially for AChE, BuChE, and COX-1 and COX-2 enzymes, suggested drug-like interactions with their active sites. The work warrants further exploration at in vivo and pre-clinical levels.

## 3. Materials and Methods

### 3.1. Chemicals

All essential oils were acquired from Doalinn, Türkiye. Cell culture materials were purchased from Gibco (San Diego, CA, USA). COX-1/COX-2 commercial screening assay kit was obtained from Cayman (Ann Arbor, MI, USA). ACE2 enzyme assay kit was purchased from BioVision (Waltham, MA, USA), and all the other chemicals including geraniol were purchased from Sigma Aldrich (St. Louis, MO, USA), with high analytical purity.

### 3.2. GC-FID and GC/MS Analyses

The Agilent 6890N GC system (Agilent 6890N GC, Santa Clara, CA, USA) was used. The simultaneous automatic injection was carried out using the same conditions in two identical columns (HP-Innowax FSC column (60 m × 0.25 mm, 0.25 μm film thickness), Agilent, Walt & Jennings Scientific, Wilmington, DE, USA) in the Agilent 5975 GC/MSD system. Relative percentages of the separated components were calculated using the FID chromatograms. For identification and characterization, in-house “Baser Library of Essential Oil Constituents” and MassFinder 3 Library (Wiley GC/MS Library, MassFinder Software 4.0) were deployed where authentic samples or the relative retention index (RRI) of *n*-alkanes was also considered [41].

### 3.3. In Vitro Studies

#### 3.3.1. COX-1/COX-2 Enzyme Inhibitory Assay

The essential oils and geraniol were tested using a commercial COX-1 (Ovine), COX-2 (Human recombinant), and fluorometric inhibitory screening assay kit following the instructions recommended by the manufacturer (Cayman test kit 700100, Cayman Chemical Company, Ann Arbor, MI, USA). Briefly, 150 µL of assay buffer (0.1 M Tris-HCl pH 8.0), 10 µL of Heme, 10 µL of enzyme (COX-1 or COX-2), and 10 µL of the test sample were added to the wells. After 5 min at 25 °C, 10 µL of ADHP (10-acetyl-3 7-dihydroxy phenoxazine) and 10 µL of arachidonic acid were added to initiate the reaction. The plate was incubated at room temperature for 2 min followed by fluorescence measured at 530 nm (excitation) and 585 nm (emission). Stock solution of the essential oils was prepared with DMSO 1%. The triplicate experimental data are expressed as mean ± standard deviation (SD) [42]. For the percentage inhibition (%I) calculations, the formula below was used:t% I = [(Initial Activity − Sample Activity)/Initial Activity] × 100

#### 3.3.2. 5-LOX Enzyme Inhibitory Assay

The essential oils and geraniol were tested using an in-house assay, where reaction was initiated by adding linoleic acid solution and the absorbance was measured at 234 nm for 10 min in a spectrophotometer. The maximum concentration of the essential oils tested was 20 μg/mL. The enzyme activity was calculated as amount of product formed per minute of the reaction for controls (without inhibitor) and test samples. Nordihydroguaiaretic acid (NDGA) was used as a positive control. All experiments were performed in triplicates and the results are reported according to standards described elsewhere [41].

#### 3.3.3. ACE 2 Enzyme Inhibitory Assay

The ACE2 enzyme inhibition assay was performed using the commercially available kit (BioVision, UK, Catalog number: K310). Stock solutions of the test samples were prepared in DMSO (1%, *v*/*v*). Essential oils (20 μg/mL) and geraniol (5 μg/mL) were dispensed into a 96-well plate followed by addition of ACE2 enzyme solution (blank excluded). The reaction was started by adding 40 µL of substrate solution to each well. The reaction mixture was measured at Ex/Em = 320/420 nm wavelength using the SpectraMax i3, fluorescent microplate reader (Molecular Devices, CA, USA). The results are reported as % inhibition values, which were obtained for all samples from triplicate experiments.

#### 3.3.4. Anticholinesterase Activity

Cholinesterase inhibition was determined spectrophotometrically through the modified Ellman method [43]. Ethanol was used as negative control. Sodium phosphate buffer, AChE enzyme stock solution, the essential oils, and geraniol were mixed and allowed to incubate for 30 min at room temperature DTNB and acetylthiocholine were added to initiate the enzyme reaction.

Butyrylthiocholine chloride was used as substrate to assay the BuChE enzyme, under the same conditions. The reaction was monitored by measuring the formation of the yellow 5-thio-2-nitrobenzoate anion at 412 nm. All experiments were performed in triplicate, and the results are reported as mean percent inhibition values.

### 3.4. Cell Culture and Conditions

Human breast cancer cell lines MCF7 (ATCC HTB-22), human prostate cancer cell lines PC3 (ATCC CRL-1435), human lung cancer cell lines A549 (ATCC CCL-185), and human embryonic kidney cell lines HEK293 (ATCC CRL-1573) were cultured in Dulbecco’s Modified Eagle’s Medium/High Glucose (DMEM, High Glucose) or Dulbecco’s Modified Eagle’s Medium/F12 (DMEM, F12) at 37 °C in a humidified 5% CO_2_ atmosphere [42]. The media were supplemented with 10% fetal bovine serum (FBS), 1% (*v*/*v*) antibiotic-antimycotics solution (100 U/mL of penicillin, 100 μg/mL of streptomycin, and 0.25 μg/mL of amphotericin B), and 1% (*v*/*v*) non-essential amino acids. 

#### Cytotoxicity Assay

Cytotoxicity activities of *R. damascena* and *P. graveolens* essential oils and geraniol were determined through cell proliferation analysis using the standard MTT assay. The in vitro cytotoxic activity of *R. damascena* and *P. graveolens* essential oils on the HEK293, A549, MCF7, and PC3 cells were assayed at a range of concentrations (0.1–2000 µg/mL). Cells were cultured for 24 h in 96-well plates containing 10^5^ cells per well before exposure. The essential oils and geraniol were dissolved in DMSO (<1%) and diluted with respective medium for use in MTT assay (0.1–2000 µg/mL concentration range). After incubation, the cells were removed and washed with D-PBS followed by addition of MTT (30 µL). After a 4 h incubation period, 150 µL of DMSO was added and mixed using a shaker at 25 °C. This was followed by measurement of the absorbances a 540 nm using the microplate reader. All experiments were performed in triplicate [42].

### 3.5. Statistical Analysis

The statistical analysis was carried out using GraphPad Prism, Version 7.02 (La Jolla, CA, USA). In vitro data are expressed as mean ± standard deviation (Mean ± SD). The *p* < 0.05 was accepted as statistically significant.

### 3.6. In Silico Assays

Analysis of each binding pose was conducted for binding score and presence of molecular interactions between geraniol and the target enzyme. Geraniol was subjected to molecular docking with all target enzymes using AutoDock Vina (v. 1.2.0) embedded in UCSF Chimera 1.14, build 42094 (University of San Francisco, USA) [44]. The crystal structures of human ACE2 (1R4L) [36], electric eel AChE (1C2B) [45], human BuChE (7AIY) [38], ovine COX-1 (2OYE) [26], human COX-2 (5IKR) [27], and soybean 5-LOX (1N8Q) [46] enzymes were obtained from the PDB server. The dock prep function of UCSF Chimera was utilized to prepare each protein for docking that included omission of all non-standard residues and addition of hydrogens and Gasteiger charges. The build structure function was used to construct the molecular structure of geraniol. Forcefield calculations were applied to analyze the ligand for hydrogen atoms, applicable charges, and rotatable bonds, followed by energy minimization. The grid area was defined for each enzyme that encompassed the whole enzyme structure. The docking procedure was conducted using the default protocol in AutoDock Vina that generated several binding poses of geraniol for each target enzyme. The top-ranked poses of geraniol with low binding energies were found to be within the active site of each enzyme. The selected poses were visually inspected and compared with known crystal structures of each enzyme in complex with their respective inhibitors for reference purposes. The PLIP server was used to verify all possible molecular interactions of geraniol with each target enzyme [47], and all figures were generated using UCSF Chimera software (v1.14, build 42094).

## 4. Conclusions

Geraniol-rich essential oils with known quality exhibit new biological and pharmacological activities that carry a utilization potential other than aromatherapy applications. In silico molecular docking studies using essential oils and their major constituents targeting a number of clinically important enzymes may lead to the discovery of new drug leads with potential treatment of various diseases.

Although in vitro and in silico methods are important tools for the evaluation of biological activity, in vivo assays and pre-clinical studies still carry high importance, establishing the safety and efficacy of the leads.

## Figures and Tables

**Figure 1 plants-12-03296-f001:**
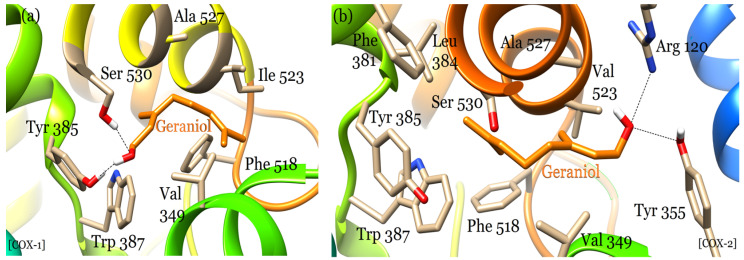
Comparison of the binding modes of geraniol for (**a**) COX-1 and (**b**) COX-2 enzymes.

**Figure 2 plants-12-03296-f002:**
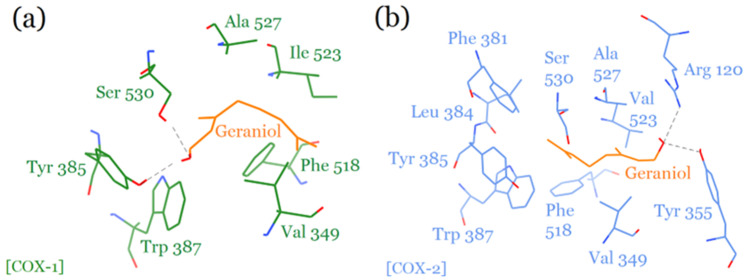
Comparison of the binding modes of geraniol on COX-1 (**a**) and COX-2 (**b**) enzymes.

**Figure 3 plants-12-03296-f003:**
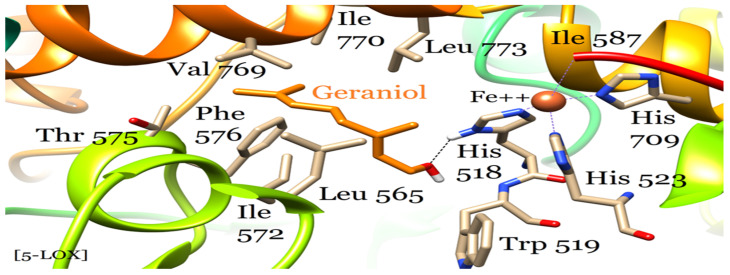
5-LOX-geraniol molecular interactions involving the active site cavity and the catalytic zinc center.

**Figure 4 plants-12-03296-f004:**
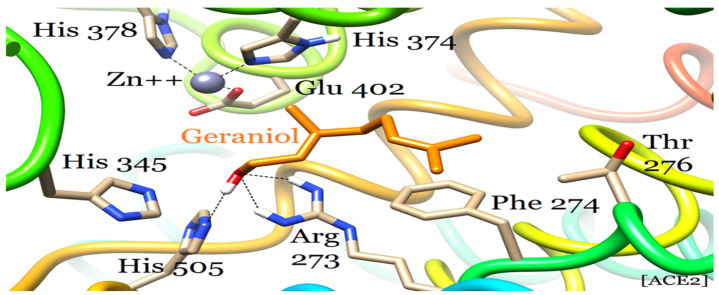
ACE2–geraniol docking complex in the S1 site of the subdomain II. The zinc center in the background is in the subdomain I of the enzyme. The hydrogen bonds in all figures are represented as black dotted lines.

**Figure 5 plants-12-03296-f005:**
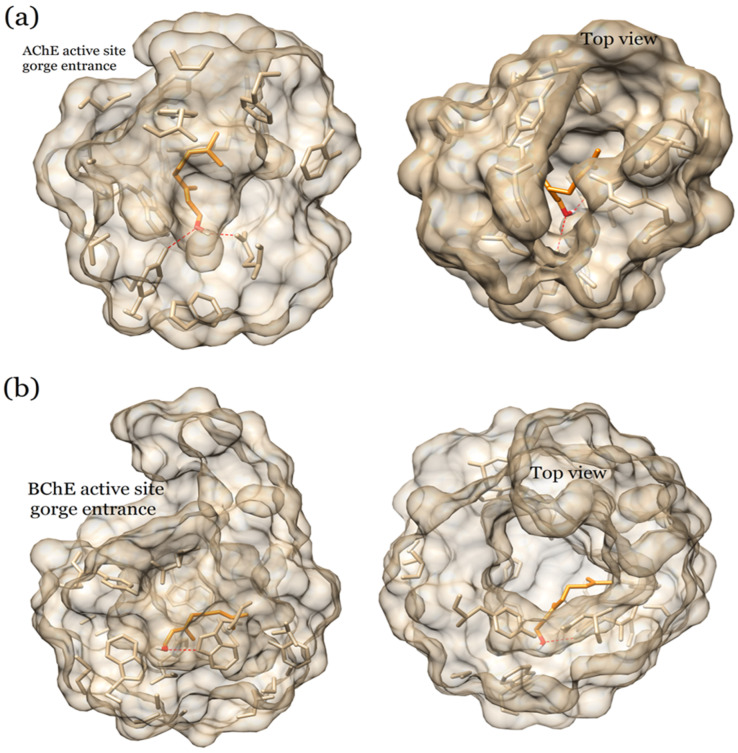
Structure of the active site gorges of AChE and BuChE. (**a**) In the AChE gorge, geraniol interacts with several important amino acids including the catalytic Glu 202 residue. (**b**) The interaction of geraniol in the BuChE gorge involves fewer amino acids than for AChE.

**Figure 6 plants-12-03296-f006:**
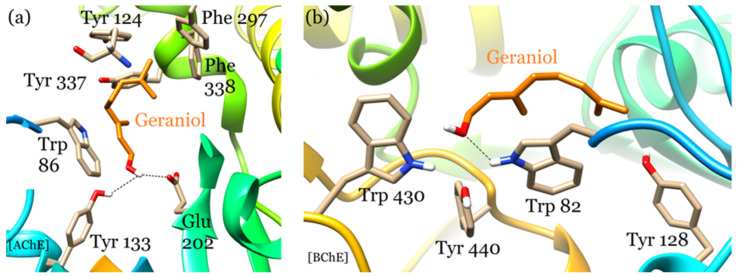
Geraniol binds in the cyclooxygenase channel of COX-1 and COX-2. (**a**) The COX-1-geraniol binding mode featuring molecular interactions with the key active site residues including the catalytic Tyr 385 and Ser 530 residues. (**b**) The COX-2-geraniol binding mode displayed a different orientation that allowed molecular interactions with active site residues other than the catalytic ones.

**Table 1 plants-12-03296-t001:** Chemical characterization of *R. damacena* (Rd) and *P. graveolens* (Pg) essential oils through GC-FID and GC/MS analyses.

RRI ^a^	Compound	Rd (%)	Pg (%)	IM
1032	α-Pinene	0.7	0.4	*t*_R_, MS
1118	β-Pinene	-	0.1	*t*_R_, MS
1174	Myrcene	-	tr	*t*_R_, MS
1203	Limonene	-	0.1	*t*_R_, MS
1280	*p*-Cymene	-	0.1	*t*_R_, MS
1362	*cis*-Rose oxide	-	0.2	MS
1376	*trans*-Rose oxide	-	0.1	MS
1475	Menthone	-	0.9	*t*_R_, MS
1503	Isomenthone	-	5.6	MS
1535	β-Bourbonene	-	0.9	MS
1553	Linalool	-	5.4	*t*_R_, MS
1579	Terpinen-4-ol	tr	-	*t*_R_, MS
1596	α-Guaiene	0.3	-	
1651	Citronellyl acetate	tr	6.3	MS
1665	Citronellyl isobutyrate	-	0.3	
1694	Neral	-	0.6	MS
1715	Geranyl formate	-	3.2	*t*_R_, MS
1726	Germacrene D	2.2	0.4	MS
1742	Geranial	tr	0.9	MS
1765	Geranyl acetate	1.3	0.4	*t*_R_, MS
1772	Citronellol	23.5	19.3	*t*_R_, MS
1773	γ-Cadinene	-	0.9	MS
1800	Octadecane	0.9	-	*t*_R_, MS
1803	Nerol	10	0.4	*t*_R_, MS
1809	Citronellyl butyrate	-	0.5	MS
1819	Geranyl isobutyrate	-	0.6	MS
1849	Calamenene	-	0.5	*t*_R_, MS
1857	Geraniol	27.3	38.7	*t*_R_, MS
1900	Nonadecane	16.5	-	*t*_R_, MS
1901	Geranyl butyrate	-	1.1	MS
1915	Nonadecene	2.4	-	*t*_R_, MS
2000	Eicosane	1.4	-	*t*_R_, MS
2000	Citronellyl hexanoate	-	0.2	*t*_R_, MS
2008	Caryophyllene oxide	-	0.3	*t*_R_, MS
2030	Methyl eugenol	1.8	-	MS
2080	Cubenol	-	0.3	MS
2100	Heneicosane	6.8	-	MS
2127	10-epi-δ-eudesmol	-	5.7	MS
2144	Spathulenol	-	0.4	*t*_R_, MS
2186	Eugenol	0.3	-	MS
2200	Docosane	1.6	-	MS
2237	Valerianol	-	0.3	MS
2287	(2*Z*,6*Z*)-Farnesol	1.6	-	*t*_R_, MS
	Total	98.9	96.0	

^a^ RRI: relative retention index calculated against n-alkanes; % calculated from FID data; tr: trace (<0.1%); IM: identification method, *t*_R_: identification based on the retention times of genuine compounds on the HP Innowax column; MS: identified on the basis of computer matching of the mass spectra with those of the Wiley and MassFinder libraries and comparison with literature data.

**Table 2 plants-12-03296-t002:** The geraniol binding energies for the target enzymes.

Enzyme	Binding Energy (Kcal/mol.)
ACE2	−5.10
AchE	−6.20
BuChE	−5.60
COX-1	−6.00
COX-2	−5.70
LOX-1	−4.40

**Table 3 plants-12-03296-t003:** Cyclooxygenase (COX) and 5-lipoxygenase (5-LOX) inhibitory activities.

	IC_50_ (µg/mL)		SI	IC_50_ (µg/mL)
	COX-1	COX-2		5-LOX
*R. damascena*	24.48	39.11	1.59	19.1
*P. graveolens*	11.13	>50	ND	23.1
Geraniol	11.17	8.11	0.72	7
Nordihydroguaretic Acid	7.1	9.1	1.28	3.6

SI: Selectivity index*;* ND: Not detected.

**Table 4 plants-12-03296-t004:** AChE and BuChE (IC_50_, µg/mL) activity of essential oils and geraniol.

	IC_50_ (µg/mL)	
	AChE	BuChE
*R. damascena*	69.39 ± 2.07	55.13 ± 1.62
*P. graveolens*	47.70 ± 1.45	50.03 ± 1.02
Geraniol	16.78 ± 1.13	9.11 ± 0.94
Galantamine (std)	11.13 ± 0.87	23.01 ± 1.07

**Table 5 plants-12-03296-t005:** Cytotoxicity evaluation of essential oils and geraniol for 24 h (IC_50_, µg/mL).

	IC_50_ (µg/mL)			
	A549	PC3	MCF7	HEK293
*R. damascena*	473.45 ± 3.11	412.18 ± 2.86	105.61 ± 1.99	ND *
*P. graveolens*	382.40 ± 2.98	112.15 ± 3.10	89.57 ± 1.58	611.82 ± 3.25
Geraniol	58.90 ± 1.09	77.02 ± 2.06	42.83 ± 1.48	81.82 ± 1.98

* ND: Not detected.

## Data Availability

Not applicable.

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
