# Peer review of "Comparative In Vitro and In Silico Enzyme Inhibitory Screening of Rosa x damascena and Pelargonium graveolens Essential Oils and Geraniol"

_plants, 2023, doi:10.3390/plants12183296_

Round 1

Reviewer 1 Report

These are my main comments on the manuscript (Plants-2573451) entitled “Comparative in vitro and in silico enzyme inhibitory screening of Rosa x damascena and Pelargonium graveolens essential oils and geraniol”. This work investigates the biological activity of R. damascena and P. graveolens essential oils, and geraniol. Following substantial revisions should be incorporated in the manuscript prior to acceptance.

1. I have concerns about the manuscript sections that I believe need to be addressed in order to improve its clarity.

2. A hypothesis for this study is needed.

3. Results and discussion should be divided in two sections, without this the manuscript cannot published.

4. In results, chemical analysis for plant metabolites needs to be checked by literature: Insert RI from literature used for identifying the compounds. Discuss mass/charge ratio in the text and (%) compound data should be in mean ± SD (I presume that the data was obtained in triplicate).

5. For each experiment, statistical method is missing and should be detailed in methods section

6. Conclusions of this study are missing.

7. Other revisions could be checked in PDF attached.

Moderate editing of English language required

Author Response

Reviewer(s)' Comments to Author:

Reviewer: 1

These are my main comments on the manuscript (Plants-2573451) entitled “Comparative in vitro and in silico enzyme inhibitory screening of Rosa x damascena and Pelargonium graveolens essential oils and geraniol”. This work investigates the biological activity of R. damascena and P. graveolens essential oils, and geraniol. Following substantial revisions should be incorporated in the manuscript prior to acceptance.

  1. I have concerns about the manuscript sections that I believe need to be addressed in order to improve its clarity.

Answer: The confusion in the titles was thoroughly revised. Thank you for your suggestion.

  1. A hypothesis for this study is needed.

Answer: “Therefore, the hypothesis of this current work is to confirm the safe traditional uses of phytochemically similar essential oils, and the major constituent geraniol using in vitro assays in correlation with in silico methods with mechanistical insights.”... was stated added the end, in the final pagaraph of the introduction.

  1. Results and discussion should be divided in two sections, without this the manuscript cannot published.

Answer: The journal rules accept “results and discussion” sections together. We condisered this as optional and decided to merge. As it was more advantageous for the flow and integrity of the data. Therefore, these sections were not separated after revisions. FYI There are some recent publications with “Results and Discussion” in Plants publications, such as, we followed:

https://www.mdpi.com/2223-7747/12/15/2859

https://www.mdpi.com/2223-7747/12/15/2847

https://www.mdpi.com/journal/plants/instructions

  1. In results, chemical analysis for plant metabolites needs to be checked by literature: Insert RI from literature used for identifying the compounds. Discuss mass/charge ratio in the text and (%) compound data should be in mean ± SD (I presume that the data was obtained in triplicate).

Answer: The work presented herein is not an analytical essential oil work, where we more focus on the biological activity of the plant preparation, moreover the major secondary metabolite geranoil. Thus we did not focus on the chemical composition of the well known Rose and Geranium oils. To ensure the quality repeatable GC-FID and GC/MS was performed as usual practise.

See for more details the published data by Betul Demirci

https://scholar.google.com/citations?user=k05W0HAAAAAJ&hl=en

  1. For each experiment, statistical method is missing and should be detailed in methods section

Answer: Thank your for your advice, the corrections were applied.

  1. Conclusions of this study are missing.

Answer: In the instructions of authors, it was stated that conclusion section is optional. Because of that we wrote the conclusion without. However, Now, we seperated this paragraph as a seerate caption.

  1. Other revisions could be checked in PDF attached.

Answer: Thank you !!! All corrections were done

--------------------------------------------------

Reviewer 2 Report

The manuscript is interesting. However, there are aspects that need to be improved. On the one hand, the introduction is very poor and includes the use of very few references (only 5) and is very outdated. On the other hand, despite the fact that figures and tables are presented in the results section, the description of the results in the text is very scarce and should be improved. In the case of the discussion it is the same, it must be improved. The manuscript does not present conclusions.

Specific comments:

1. The abstract must not include abbreviations (for example, MTT, COX-1, COX-2, etc). They must be defined the first time they are mentioned.

2. The abstract must be rewritten. It mainly focuses on displaying the results in an unordered manner.

3. For the introduction, I suggest incorporating information regarding the context of the study, the characterization and use of the plants, as well as the importance of the enzymatic activities used at a physiological and health level. All this using reference ideally from the last 5 years.

4. The hypothesis must be added at the end of the introduction.

5. Lines 68 and 69: The values (2.5 and 20%) do not coincide with those reported in the table.

6. Table 1: What number of replicates were made for these determinations? Each result must be associated with its error (standard error, standard deviation or similar). The analytical validation was carried out?

7. Point 2.2: The description of the results and discussion are missing.

8. Line 113: 638.79… Where did this value come from?

9. Lines 117-118: Please rewrite. Missing a value?

10. Table 3: ND?

11. Point 2.2-2.4: Please, improve discussion

12. Table 4: Please, define ALL abbreviations

13. Point 3.2: Please add the city and country of the GC system manufacturer and the references of the analysis conditions

14. Line 373: 105?

15. Missing conclusions

Author Response

Reviewer: 2

The manuscript is interesting. However, there are aspects that need to be improved. On the one hand, the introduction is very poor and includes the use of very few references (only 5) and is very outdated. On the other hand, despite the fact that figures and tables are presented in the results section, the description of the results in the text is very scarce and should be improved. In the case of the discussion it is the same, it must be improved. The manuscript does not present conclusions.

Specific comments:

  1. The abstract must not include abbreviations (for example, MTT, COX-1, COX-2, etc). They must be defined the first time they are mentioned.

Answer: The abbreviations were deleted from abstract.

  1. The abstract must be rewritten. It mainly focuses on displaying the results in an unordered manner.

Answer: The abstract was  rewritten.

  1. For the introduction, I suggest incorporating information regarding the context of the study, the characterization and use of the plants, as well as the importance of the enzymatic activities used at a physiological and health level. All this using reference ideally from the last 5 years.

Answer: The introduction section was improved and extended as suggested.

  1. The hypothesis must be added at the end of the introduction.

Answer: The hypothesis was added

  1. Lines 68 and 69: The values (2.5 and 20%) do not coincide with those reported in the table.

Answer: It is corrected, Table values were correct.

  1. Table 1: What number of replicates were made for these determinations? Each result must be associated with its error (standard error, standard deviation or similar). The analytical validation was carried out?

Answer:  As the commercial oils are well known and authentic, no detailed essential oil analysis was performed. However, simultaneous GC-FID and GC/MS analyses ensure, that the obtained data is valid. Also the data obtained is correlated to literature. The main aim of this work was to demonstrate the importance of Rose and Pelargonium oils based on geraniol were in vitro results were supported by in silico data. Also new in vitro tox data was obtained. Thus there is the basic chemical information needed provided.

  1. Point 2.2: The description of the results and discussion are missing.

Answer: Thank you – was corrected.

  1. Line 113: 638.79… Where did this value come from?

Answer: It is corrected.

  1. Lines 117-118: Please rewrite. Missing a value?

Answer: It is completed.

  1. Table 3: ND?

Answer: It is added. Not detected!

  1. Point 2.2-2.4: Please, improve discussion

Answer: Discussion section was improved.

  1. Table 4: Please, define ALL abbreviations

Answer: All were carefully evaluated and defined/ explained whereever needed

  1. Point 3.2: Please add the city and country of the GC system manufacturer and the references of the analysis conditions

Answer: The information was added.

  1. Line 373: 105?

Answer: It is corrected.

  1. Missing conclusions

Answer: In the instructions to authors, it was stated that conclusion section is optional.

HOwever, this work deserved a conclusion which we added for the audience.

Thank you!

Round 2

Reviewer 1 Report

The authors have incorporated all suggestions and reviewer comments into the latest version, now the manuscript seems much clear. There are minor points to be corrected:

L.28: …gas chromatography – flame ionization detector (GC-FID) as...

Ls.65-55: Revise this sentence to eliminate rewordiness.

Ls.91: …mammalian cholinesterase…

Ls. 104-107: Place this sentence before lines 101-104 (before main objective).

No comments.

Author Response

First of all, the authors would like to thank the reviewers and editor for their constructive comments as well as their extensive valuable suggestions. Below are the point-by-point responses, where the changes in the revised manuscript were also colored highlighted.

Reviewer(s)' Comments to Author:

Reviewer: 1

The authors have incorporated all suggestions and reviewer comments into the latest version, now the manuscript seems much clear. There are minor points to be corrected:

L.28: …gas chromatography – flame ionization detector (GC-FID) as...

Ls.65-55: Revise this sentence to eliminate rewordiness.

Ls.91: …mammalian cholinesterase…

Ls. 104-107: Place this sentence before lines 101-104 (before main objective).

Answer: All corrections were done. Thank you for the constructive review.

Reviewer 2 Report

Despite the fact that the manuscript has been improved, there are still aspects that need to be reviewed and improved.

1. At the end of the abstract, a conclusion/projections paragraph should be incorporated.

2. Lines 53 and 57: For each of these paragraphs only one reference was used. Please, incorporate all the references that are necessary to support the information.

3. Lines 65 and 80: Missing references.

4. Table 1: What number of replicates were made for these determinations? Each result must be associated with its error (standard error, standard deviation or similar). The analytical validation was carried out?. It is of great importance to incorporate this information.

5. Point 2.2: The description of the results and discussion are missing: This section is pending improvement.

6. Point 2.2-2.4: Please, improve the discussion section

7. Table 4 is missing in the manuscript.

Author Response

First of all, the authors would like to thank the reviewers and editor for their constructive comments as well as their extensive valuable suggestions. Below are the point-by-point responses, where the changes in the revised manuscript were also colored highlighted.

Reviewer: 2

Despite the fact that the manuscript has been improved, there are still aspects that need to be reviewed and improved.

  1. At the end of the abstract, a conclusion/projections paragraph should be incorporated.

Answer: A very limited conclusion  could be added as there is    200 word limit of PLANTS.

  1. Lines 53 and 57: For each of these paragraphs only one reference was used. Please, incorporate all the references that are necessary to support the information.

Answer: Yes, the reference 4 is a review article, which covers the statement.

  1. Lines 65 and 80: Missing references.

Answer: The cited Reference 8 is also a review article on P. graveolens. And all the info is included there. Also, Ref 13 contains information on the citation.

  1. Table 1: What number of replicates were made for these determinations? Each result must be associated with its error (standard error, standard deviation or similar). The analytical validation was carried out?. It is of great importance to incorporate this information.

Answer: As stated previously, the work core is not an analytical evaluation of essential oils. Morevoer, both commercial essential oils are wellknown, which comply according the GC-FID and GC/MS analyses with current accepted literature. This is not a quantitative and qualitative analytical essential oil analysis study to confirm. Our methods are stable and repeatable in terms of essential oil analyes. If the referee insists we can indeed sent him the oils for evaluation and confirmation. Also, these oils are commercially avaliable. Our group has hundreds of essential oil analysis publications in Journal of Essential Oils, Flavor and Frangrances Journal, Journal of Agricultural Food Chemistry, Food Chemistry, Planta Medica, etc among hundereds of other well reputed journals in Essential Oil analyses, also for authentic and endemic essential oils from minute amounts including quantitative analyses, too.

This present study does not require validation study with quantitative analyses, as the work focuses on new in vitro enzyme inhibition studies supported by in silico simulations for the first time on Geraniol and the oils containing it in relatively high amounts for the very first time.

  1. Point 2.2: The description of the results and discussion are missing: This section is pending improvement.

Answer: Actually, 2.2. is not a seperate heading. We revised and replaced it in the correct place in this present revision.

  1. Point 2.2-2.4: Please, improve the discussion section

Answer: The discussion section was extended. However, since ACE2 enzyme inhibition studies are a relatively new field, and we have published several essential oil data also on this topic, citations to relevant journals were made.

  1. Table 4 is missing in the manuscript.

Answer: Table 4 was Corrected. Thank you for your careful evaluation and support.

Round 3

Reviewer 2 Report

Regarding manuscript 2573451-v3. Although some of the comments were incorporated into the text, there are still areas for improvement.

1. At the end of the abstract, a conclusion/projections paragraph should be incorporated: It is understood that the limit allowed by the journal is 200 words. However, a section of great importance in the abstract are the conclusions and projections, for which I suggest reducing, as much as possible, some of the other sections to include what is suggested.

2. Lines 53 and 57: For each of these paragraphs only one reference was used. Please, incorporate all the references that are necessary to support the information: In cases like this, it is recommended to include the references of those who carried out the experimental work of those who studied the reported effect, not of a review.

3. Table 1: What number of replicates were made for these determinations? Each result must be associated with its error (standard error, standard deviation or similar). The analytical validation was carried out?. It is of great importance to incorporate this information: Although the authors indicate that no quantitative determinations were made in the oils, what is reported in table 1 and in the methodology section 3.2, corresponds to a quantitative analysis where the concentration, using the % concentration unit, of different components of the essential oils. To carry out any quantitative determination, it is necessary to carry out its prior analytical validation, both at the level of the optimization of the analysis methodology, and for the calibration of the analysis of the samples. Otherwise, it is not possible to report reliable results.

Author Response

Mans: plants-2573451  - 3. revision date: 1.9.23

Comments and answers to the reviewers

First of all, the authors would like to thank the reviewers and editor for their constructive comments as well as their extensive valuable suggestions. Below are the point-by-point responses, where the changes in the revised manuscript were highlighted.

Reviewer(s)' Comments to Author:

  1. At the end of the abstract, a conclusion/projections paragraph should be incorporated: It is understood that the limit allowed by the journal is 200 words. However, a section of great importance in the abstract are the conclusions and projections, for which I suggest reducing, as much as possible, some of the other sections to include what is suggested.

Answer: As per as the suggestion of the reviewer, the authors added a conclusion sentence to the abstract, forcing the 247 words, despite the limits.

In the abstract, the journal wants us to talk about both the backround, the method and the results. It was difficult for a comprehensive study to express all these together with a limited number of words. However, minor additions were made to both the abstract and the text.

“In conclusion, initial promising bioactivity studies suggest further complementary in vivo studies to confirm the safe use of essential oils.”

 Also within the manuscript in the 4. Conclusion part was expanded with future actions to be done

“Geraniol rich essential oils with known quality have still new biological and pharmacological activity and utilization potential, not only for aromatherapeutical applications. It can be justified that still detailed research is needed, both in terms of activity and toxicity evaluations. In silico molecular docking studies using essential oils and their major constituents targeting special enzymes may lead to the discovery of new drugs or leads with potential entities for the treatment of various diseases after toxicological considerations.

Although in vitro and in silico methods are an important tool for evaluation of biological activity, in vivo assays, and moreover clinical studies have still high importance and are obligatory for confirming the tentative experimental data for safety and efficacy”

Infact, for an abstract, the material and method part is actually more important, to show reality. Conclusion is more of a imagination without limits. Not to forget, this is an openaccess publication, so the reader is not just limited to the abstract, which may be the case for hundred thousands of publications where only DATA from the abstract can be retrived, thus the Conclusion is not vital!

  1. Lines 53 and 57: For each of these paragraphs only one reference was used. Please, incorporate all the references that are necessary to support the information: In cases like this, it is recommended to include the references of those who carried out the experimental work of those who studied the reported effect, not of a review.

 Answer:

As it was in the introduction and to keep it short, the recent reviews on the topic were initially cited. However, to help the reader each primary reference was cited in the applicable parts as per suggestion. The citations increased also in number for this research work.  A NEW research work should not have more than 40 citations, as it will decrease the orginaltiy! The aim of the present work was just to cite experimentally relevant literature, rather to write in a shortened review fashion. Nevertheless, in the revised form the  relevant citations were added.

  1. Table 1: What number of replicates were made for these determinations? Each result must be associated with its error (standard error, standard deviation or similar). The analytical validation was carried out?. It is of great importance to incorporate this information: Although the authors indicate that no quantitative determinations were made in the oils, what is reported in table 1 and in the methodology section 3.2, corresponds to a quantitative analysis where the concentration, using the % concentration unit, of different components of the essential oils. To carry out any quantitative determination, it is necessary to carry out its prior analytical validation, both at the level of the optimization of the analysis methodology, and for the calibration of the analysis of the samples. Otherwise, it is not possible to report reliable results.

Answer: In the present work, as reported, 2 repeatable solid GC-FID and GC/MS analyses were performed. In quantitative sample analyses your suggestions are applicable and a must, however, in qualitative analyses our analyses are accepted in reputed essential oil journals. IF you carefully read the table footnote as below, you will follow that we do NOT report concentrations nor QUANTITATIVE component results. Therefore, no further validation efforts are needed in the present work.

The method of calculation and identification was briefly mentioned in line 125 below table 1.

aRRI: Relative retention indices calculated against n-alkanes; % calculated from FID data; tr: Trace (< 0.1 %);  IM: Identification method, tR, identification based on the retention times of genuine compounds on the HP Innowax column; MS, identified on the basis of computer matching of the mass spectra with those of the Wiley and MassFinder libraries and comparison with literature data

in the experimental part we have described the routine analytical qualitative method used in more than 500 publications, since 1991.

https://www.khcbaser.com/publications/publications-in-international-refereed-journals/

As already explained in the previous communications in answers1 and 2, this present work is NOT an essential oil analysis report, however, reliable and repeatable GC-FID and GC/MS qualitative relative % of identified components were also reported in this present study for very well known Rose and Geranium essential oils comparable with the current known literature and market specifications. Our results moreover, focus on the main component geraniol on in vitro and in silico enzyme inhibition results.

As a conclusion, the essential oil analytical results are acceptable in the present reported form.

Round 4

Reviewer 2 Report

No comments.